# FewViewGS: Gaussian Splatting with Few View Matching and Multi-stage Training

**Ruihong Yin**
University of Amsterdam
r.yin@uva.nl

**Vladimir Yugay**
University of Amsterdam
v.yugay@uva.nl

**Yue Li**
University of Amsterdam
y.li6@uva.nl

**Sezer Karaoglu**
University of Amsterdam
3DUniversum
s.karaoglu@3duniversum.com

**Theo Gevers**
University of Amsterdam
3DUniversum
th.gevers@uva.nl

## Abstract

The field of novel view synthesis from images has seen rapid advancements with the introduction of Neural Radiance Fields (NeRF) and more recently with 3D Gaussian Splatting. Gaussian Splatting became widely adopted due to its efficiency and ability to render novel views accurately. While Gaussian Splatting performs well when a sufficient amount of training images are available, its unstructured explicit representation tends to overfit in scenarios with sparse input images, resulting in poor rendering performance. To address this, we present a 3D Gaussian-based novel view synthesis method using sparse input images that can accurately render the scene from the viewpoints not covered by the training images. We propose a multi-stage training scheme with matching-based consistency constraints imposed on the novel views without relying on pre-trained depth estimation or diffusion models. This is achieved by using the matches of the available training images to supervise the generation of the novel views sampled between the training frames with color, geometry, and semantic losses. In addition, we introduce a locality preserving regularization for 3D Gaussians which removes rendering artifacts by preserving the local color structure of the scene. Evaluation on synthetic and real-world datasets demonstrates competitive or superior performance of our method in few-shot novel view synthesis compared to existing state-of-the-art methods.

## 1 Introduction

Reconstruction of a 3D scene representation from sparse 2D observations that can render novel views unseen during training has been an active field of research with wide applications in VR/AR and navigation. Recently, neural radiance fields (NeRF) [20] utilizing differentiable volume rendering and implicit scene representation achieved great results in novel view synthesis (NVS). The NeRF framework was extended to the sparse-view setting (few-shot NVS) using efficient training schemes, regularization losses, depth consistency, and image priors [11, 22, 8, 34]. While being good at rendering, NeRFs typically take a long time to be optimized, and their rendering speed is far from real-time which greatly limits their practical application.

3D Gaussian Splatting [13] has introduced an unstructured radiance field represented with 3D Gaussians. Gaussians are initialized from a sparse Structure-from-Motion (SfM) [29] point cloud and dynamically added or removed from the scene during training. With images observing a static scene from different viewpoints, Gaussian parameters are optimized using photometric loss. This method became widely adopted due to its training efficiency, unprecedented rendering speed, and quality. However, the rendering performance dramatically drops when fewer images are used for

training. While well-observed regions can be accurately rendered, less-supervised scene geometry is under-reconstructed, Gaussian parameters are overfitted to the views observing the scene [44], geometry artifacts [38] appear. In addition, optimization is highly dependent on the SfM initialization point cloud whose quality is also affected by the sparsity of the training views. There are several concurrent works exploring few-shot Gaussian splatting [15, 38, 24, 6, 46]. However, all of them rely on depth estimation [15, 38, 24, 6, 46] or diffusion priors [38]. In these methods, depth-estimation networks predict depth up to scale on unseen scenes, making them sensitive to domain shifts, while diffusion models increase training time.

In this paper, we introduce **FewViewGS**, a Gaussian splatting-based novel view synthesis method achieving state-of-the-art rendering results in a few-shot setup. We break the training process into the pre-training, intermediate, and tuning stages to better propagate the information to the novel views. During the pre-training stage, we use only training views to get a basic representation of the scene. This enables us to obtain a fundamental point cloud and scaled training view depth maps. In the intermediate stage, the emphasis is on optimizing the new perspectives. We use multi-view geometry and a novel view interpolation sampling to render unseen views coherent with the training images. For this, we match the pairs of training images, robustly warp the matches to the randomly sampled novel views between them, and enforce novel view consistency by applying geometry, color, and semantic losses. In the final stage, the scene representation undergoes refinement through a limited number of iterations, utilizing only the known views. During training, the 3D Gaussians are regularized by our proposed locality preserving regularization to maintain their local properties, eliminating the artifacts on the novel views. In summary, our **contributions** are as follows:

- A few-shot NVS Gaussian splatting-based system not relying on pre-trained depth estimation or diffusion models achieving SoTA rendering results.
- A multi-stage training scheme enabling seamless knowledge transfer from known to novel views.
- A robust warping-based novel view consistency constraint ensuring the coherence of the synthesized unseen images.
- 3D Gaussian locality preserving regularization handling visual artifacts.

## 2 Related work

**Novel View Synthesis.** Neural Radiance Fields [20] model a 3D scene using a neural network that takes the viewing direction and 3D location as input and predicts color and density. During training, the network is optimized to render the images of a scene from different angles. The network trained in such a fashion implicitly learns how to interpolate and extrapolate [39] to render the scene from the angles unseen during training at a very high resolution. Due to its capabilities, NeRF became widely adopted for 3D scene reconstruction [1, 18, 8, 34, 14], human body modeling [26, 25, 36, 16], robotics [41, 28], and medical imaging [7].

However, classical NeRFs require extensive training time and computational resources, making them less practical for real-time applications. There are several methods [2, 1, 21, 10, 42] focusing on efficiency by using multiscale scene representations [1] or efficient data structures for scene encoding [21, 10, 42, 2]. However, all those approaches come at the cost of lower rendering quality.

Recently introduced, 3D Gaussian Splatting [13] uses 3D Gaussians to represent the scene. To render a view of a scene, the 3D Gaussians in the camera frustum view are splatted to the image plane. Having several training images observing a static scene from different angles, the parameters of the 3D Gaussians are optimized by minimizing the photometric loss. Due to explicit geometry representation, this method achieves real-time rendering and fast optimization without compromising rendering quality. However, just as NeRF, Gaussian splatting requires multiple views observing the scene from various angles.

**Few-shot Novel View Synthesis.** Few-shot novel view synthesis aims to train a scene representation capable of generating new views of a scene using only a sparse number of training images. Due to the lack of multi-view constraints on the scene and usage of photometric losses, training such a representation remains challenging.

Several works explore how to train better NeRFs for the novel view synthesis in a sparse setting. One group of methods explores the regularization with priors from pre-trained neural networks [8, 34, 37]. For example, DSNeRF [8] and SparseNeRF [34] use depth regularization from pre-trained depth estimators on known views to guide optimization. DiffusioNeRF [37] uses weight regularization and priors from diffusion models.

Another line of work [4, 22, 11, 33, 31] focuses on imposing additional supervision on the novel views during training. For example, GeoAug [4] randomly samples novel views around the known frames and then calculates the color loss between the warped novel view and the known view. RegNeRF [22] designs depth smooth regularization on unobserved views. DietNeRF [11] argues that the high-level semantic information should be similar for individual scenes and proposes to apply semantic consistency loss on the novel views. SPARF [33] integrates multi-view correspondence and geometry loss to the optimization. ViP-NeRF [31] regularizes the network with visibility prior, which is generated by plane sweep volumes.

Simultaneously with our research, numerous methods have emerged focusing on few-shot Gaussian splatting for novel view synthesis. FSGS [46] proposes a Gaussian Unpooling strategy to generate dense Gaussians. SparseGS [38] adopts a floater removal strategy to remove unnecessary Gaussians close to the camera. DNGaussian [15], CoherentGS [24], DRGS [6] focus on regularizing the depth maps. Remarkably, all the few-shot novel view synthesis methods based on 3DGS try to integrate the pre-trained depth estimation networks in the pipelines.

Novel view regularization has shown great results in sparse novel view synthesis. However, sampling random novel views in the camera frustum can make the rendering results inconsistent. We therefore propose to use correspondences between the known views to regularize the novel views sampled between them robustly. Moreover, we avoid relying on complex priors. Using simple image matching in the 2D-pixel space, we avoid using depth estimation models predicting depth up to scale and heavy diffusion models which struggle on out-of-distribution datasets, while achieving state-of-the-art rendering results.

## 3 Method

The key idea of our approach is to enforce consistency between the sparse known views and the novel views in visually overlapping areas. As presented in Fig. 1, we introduce a multi-stage training scheme consisting of pre-training, intermediate, and tuning stages to gradually optimize the scene and obtain depth maps for the training views. After the pre-training stage, we start enforcing consistency of the novel views. For this, we first sample random training frames and match them with each other in pixel space. Further, we randomly sample a pose between the pairs of training frames and warp the matched pixels to it. We then filter out unreliable warped pixels and apply our new color, geometry, and semantic losses to the rendered novel view. This ensures the novel view remains consistent only in the regions that overlap with the known view pairs. In addition, we apply locality-preserving regularization to remove typical artifacts that appear in few-shot scenarios. We now outline our pipeline, beginning with an overview of Gaussian Splatting [13], followed by the novel view consistency mechanism, regularization losses, and our training scheme.

### 3.1 Preliminary

Gaussian splatting [13] is a recent radiance field representation that replaces the implicit neural network with explicit optimizable 3D Gaussians.

A single 3D Gaussian is parameterized by its mean $\mu \in \mathbb{R}^3$, covariance $\Sigma \in \mathbb{R}^{3 \times 3}$, opacity $o \in \mathbb{R}$, and a set of spherical harmonics coefficients $sh \in \mathbb{R}^{3(l+1)^2}$ with degree $l$ for view-dependent color. The mean of a projected (splatted) 3D Gaussian in the 2D image plane $\mu_I$ is computed as follows:

$$\mu_I = \pi\big(P(T_{wc}\hat{\mu})\big) \ , \tag{1}$$

where $\hat{\mu}$ denotes homogeneous mean, $T_{wc} \in SE(3)$ is the world-to-camera extrinsics, $P \in \mathbb{R}^{4 \times 4}$ is an OpenGL-style projection matrix, $\pi$ is the projection to pixel coordinates. The 2D covariance $\Sigma_I$ of a splatted Gaussian is computed as:

$$\Sigma_I = JR_{wc}\Sigma R_{wc}^T J^T \ , \tag{2}$$

where $J \in \mathbb{R}^{2 \times 3}$ is the Jacobian of the affine approximation of the projective transformation, $R_{wc} \in SO(3)$ is the rotation component of $T_{wc}$. For further details on the splatting process, we refer to [47].

Parameters of the 3D Gaussians are iteratively optimized by minimizing the photometric loss between rendered and training images. During optimization, covariance is decomposed as $\Sigma = RSS^T R^T$, where $R \in \mathbb{R}^{3 \times 3}$ and $S = \text{diag}(s) \in \mathbb{R}^{3 \times 3}$ are rotation and scale respectively to preserve covariance

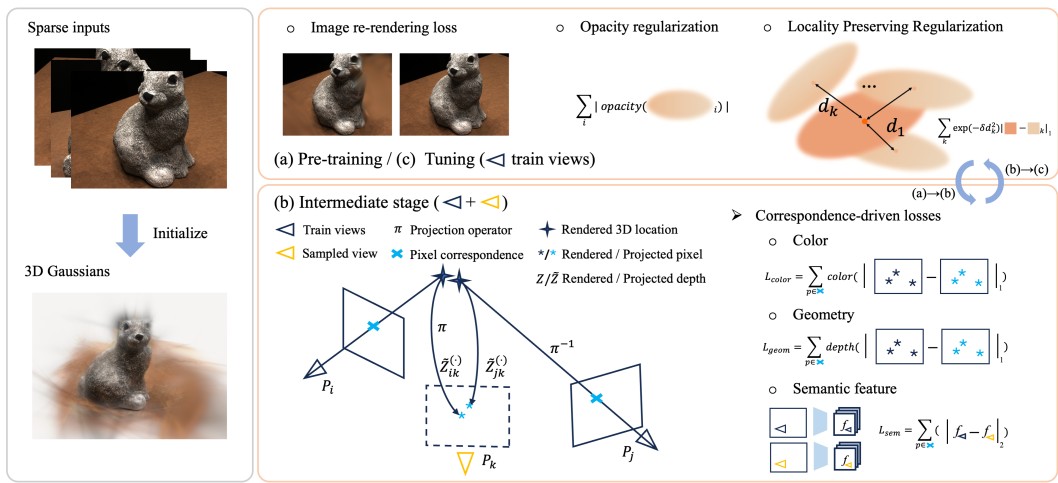

**Figure 1: FewViewGS pipeline.** Our method consists of a multi-stage training scheme of (a) pre-training, (b) intermediate, and (c) tuning stages. **Top right: pre-training / tuning.** At the beginning and end, Gaussians are optimized solely on the known input views, utilizing color re-rendering loss and regularization terms on total opacity and local appearance. **Bottom right: intermediate.** Correspondences are first extracted from the pairs of training images and projected onto the virtual sampled views. Given the projected and virtual renders, color, geometry, and semantic losses are calculated at the projected pixels in the new views.

positive semi-definite property. Color $c$ for a pixel influenced by $n$ ordered Gaussians is rendered as:

$$c = \sum_{i=1}^{n} c_i \cdot \alpha_i \cdot T_i \ , \ \text{with} \ T_i = \prod_{j=1}^{i-1}(1 - \alpha_i) \ , \tag{3}$$

with $c_i$ the color of $i^{th}$ Gaussian and $\alpha_i$ computed as:

$$\alpha_i = o_i \cdot \exp(-\sigma_i) \quad \text{and} \quad \sigma_i = \frac{1}{2}\Delta_i^T \Sigma_{I,i}^{-1} \Delta_i \ , \tag{4}$$

where $\Delta_i \in \mathbb{R}^2$ is the offset between the 2D mean of a splatted Gaussian and the pixel coordinate. Similarly, this blending process could be extended to render pixel depth. We compute the depth $d$ of a pixel that is influenced by $n$ ordered Gaussians as:

$$d = \sum_{i=1}^{n} \mu_i^z \cdot \alpha_i \cdot T_i \ , \tag{5}$$

where $\mu_i^z$ is the $z$ component of the mean , $\alpha_i$ and $T_i$ are the same as in Eq. (3).

This splatting approach provides significantly faster training and inference time with no compromise on rendering quality compared to NeRF. During training, new Gaussians are added and removed from the scene based on the heuristics introduced in [13] to improve the scene coverage.

## 3.2 Novel View Consistency

Our main goal is to ensure the novel views are consistent with the training views in overlapping regions, maintaining visual coherence. However, determining which specific pixels in the training views should match those in the novel views is challenging.

Other few-shot methods [4, 22, 11, 33, 31] try to address this by randomly sampling novel views within the viewing frustum of a training frame and then applying regularization techniques to them. In contrast, our approach uses pairs of training views to more robustly identify which pixels need regularization in the novel views. This method enhances accuracy and consistency, aligning novel views more effectively with the training data, resulting in higher quality and more coherent visual output.

As the first step, for a pair of training images $i$ and $j$ we find the pixel correspondences $C_i, C_j$ using a robust image matcher [9]. Since the training views' poses $P_i$ and $P_j$ are known, a novel view pose $P_k$ is randomly sampled between them. Further, we render the depth maps of the matched pixels $Z_i, Z_j$ of the training views, and warp them to the novel view, defined by:

$$C_{u,i} = \pi(P_k \cdot \pi^{-1}(C_i, Z_i)), \tag{6}$$

where $C_{u,i}$ are the pixels of the novel view, $\pi$ and $\pi^{-1}$ are the projections from 3D to 2D and from 2D to 3D respectively.

Having projected the matched pixels to the novel view, they are used to supervise novel view synthesis by applying color, geometric, and semantic losses. However, natural errors stemming from color and depth rendering, or image matching, make supervision over all the projected pixels unstable. To mitigate these effects, an agreement masking is proposed and defined as follows:

$$M(x_i, x_j, \theta) = I\left(|x_i^k - x_j^k| < \theta\right), \tag{7}$$

where $x_i, x_j$ are image-shape matrices, $\theta \in R$ is an agreement threshold, and $I$ is an indicator function.

In addition, the warped values often end up in the wrong locations when warped from the high color gradient regions, contaminating the losses. This happens mostly because of inaccurate matches in such areas. To address this, we introduce a color gradient weighting $W$ which alleviates the effect of errors in texture-rich regions. For a color gradient $G$ at a pixel $(u, v)$, it is computed as:

$$W(G_{u,v}, \theta_{grad}) = \begin{cases} exp(-G_{u,v}) & \text{if } |G_{u,v}| > \theta_{grad} \\ 1 & \text{else} \end{cases} \tag{8}$$

where $|G_{u,v}|$ is gradient magnitude, $\theta_{grad}$ is a scalar threshold.

Denoting matches between known images $i$ and $j$ as $P_{i,j}$, we enforce the geometric consistency of the novel view using the loss:

$$L_{geom} = \sum_{p \in P_{i,j}} M(\tilde{Z}_{ik}^p, \tilde{Z}_{jk}^p, \theta_g) \cdot W(G_t^p, \theta_{grad}) \cdot min(|Z_k^p - \tilde{Z}_{ik}^p|_1, |Z_k^p - \tilde{Z}_{jk}^p|_1), \tag{9}$$

where $Z_k^p$ is the rendered depth of a novel view $k$ at a pixel $p$, $\tilde{Z}_{ik}^p, \tilde{Z}_{jk}^p$ are the projected depth values from the known views $i$ and $j$ respectively, $\theta_g$ is an agreement threshold. $t$ is set to $i$ if $|Z_k^p - \tilde{Z}_{ik}^p|_1$ is less than $|Z_k^p - \tilde{Z}_{jk}^p|_1$ and to $j$ otherwise. We take the minimum over the pair of depth discrepancy to further reduce the influence of inaccurate projections.

To regularize the appearance of the novel view, the loss in the color space is applied as follows:

$$L_{color} = \sum_{p \in P_{i,j}} M(\tilde{Z}_{ik}^p, \tilde{Z}_{jk}^p, \theta_g) \cdot W(G_t^p, \theta_{grad}) \cdot min(|c_k^p - \tilde{c}_{ik}^p|_1, |c_k^p - \tilde{c}_{jk}^p|_1), \tag{10}$$

where $c_k^p$ is the rendered color of a novel view $k$ at pixel $p$. $t$ is defined following the same logic as in Eq. (9).

Finally, the novel view should have similar semantics to the training views in the overlapping regions. Since pre-trained deep neural networks can encode semantic information of the images [30, 27, 23], the loss is added on the feature space of the novel views:

$$L_{sem} = \sum_{p \in P_{i,j}} M(\tilde{Z}_{ik}^p, \tilde{Z}_{jk}^p, \theta_g) \cdot W(G_t^p, \theta_{grad}) \cdot min(|f_k^p - f_i^p|_2, |f_k^p - f_j^p|_2), \tag{11}$$

where $f_i, f_j, f_k$ are image-sized features extracted with a pre-trained network from the views $i, j, k$ respectively. $t$ is defined following the same logic as in Eq. (9).

The final equation for the novel view consistency loss is expressed as:

$$L_{consistency} = \alpha \cdot L_{geom} + \beta \cdot L_{color} + \gamma \cdot L_{sem}, \tag{12}$$

where $\alpha, \beta, \gamma$ are the respective scalar weights.

### 3.3 Locality Preserving Regularization

After examining optimized 3D scenes in a few-shot setting, it was found that accurate rendering results from Gaussian splatting occur when color values are smooth within local neighborhoods. However, in a standard setting, the photometric loss does not explicitly enforce this. Although this is not an issue when numerous frames are available, it leads to artifacts in a few-shot setup.

Therefore, a locality-preserving regularization is proposed to address this issue. For every Gaussian $i$, its neighborhood $N_i$ is defined by finding the $K$ nearest neighbors in 3D space. Furthermore, the color parameters of Gaussian $i$ are enforced to be proportionally closer to those of its neighborhood:

$$L_{locality} = \sum_{k \in N_i} exp(-\delta \cdot |\mu_k - \mu_i|_2) \cdot |c_k - c_i|_2, \tag{13}$$

where $\delta$ is a scaling factor, $c_i$ and $c_k$ are the colors of the Gaussian $i$ and its neighbor $k$ respectively. The weight function $exp(-\delta \cdot |\mu_k - \mu_i|_2)$ measures the influence of the neighbor colors based on their distance to the Gaussian center. The effect of locality regularization is demonstrated quantitatively in Tab. 4 and qualitatively in Tab. 2.

### 3.4 Multi-stage Training Scheme

Our training scheme serves two goals. First, to avoid overfitting to the training views and to provide a coarse initialization for synthesizing novel views. Second, to provide scaled depth estimates for the training views to allow warping.

During the pre-training stage, only the training views are used to optimize the 3D Gaussians for a small number of iterations using the loss:

$$L_{pre-training} = \lambda \cdot L_{photometric} + \chi \cdot L_{opacity} + \zeta \cdot L_{locality}, \tag{14}$$

where $L_{photometric}$ is the photometric loss from [13], $L_{opacity}$ is the $L2$ regularization term for the Gaussians' opacity, and $\lambda, \chi, \zeta$ are the scalar weights. Opacity regularization [32] is added to remove the unobserved floating Gaussians that cause floating artifacts.

In the intermediate stage, novel views are synthesized and used as supervision for the Gaussians to avoid overfitting. We found that sparse supervision from feature matching and noise in pseudo labels for novel views leads to model collapse in certain scenes. To address this, training views are also used to regularize the scene representation, with a downscaled $L_{pre-training}$. Since the depth maps become available after the pre-training stage, we start using novel view consistency losses as described in Sec. 3.2:

$$L_{intermediate} = \kappa \cdot L_{consistency} + \eta \cdot L_{pre-training}, \tag{15}$$

where $\kappa, \eta$ are the scalar weights for the respective losses.

The tuning stage is designed to refine the network using ground truth, provide supervision for unmatched pixels during the intermediate phase, and reduce the impact of noise in novel views. At the same time, it is crucial not to overfit the training views. Only training views are used for optimizing the $L_{pre-training}$ loss for a small number of iterations.

## 4 Experiments

Here we describe our experimental setup and then evaluate our approach against state-of-the-art few-shot NVS methods on commonly used datasets [12, 19, 20]. In addition, we compare our method with concurrent works. The tables highlight best results as first , second , third . Concurrent works in the tables are marked with an asterisk*.

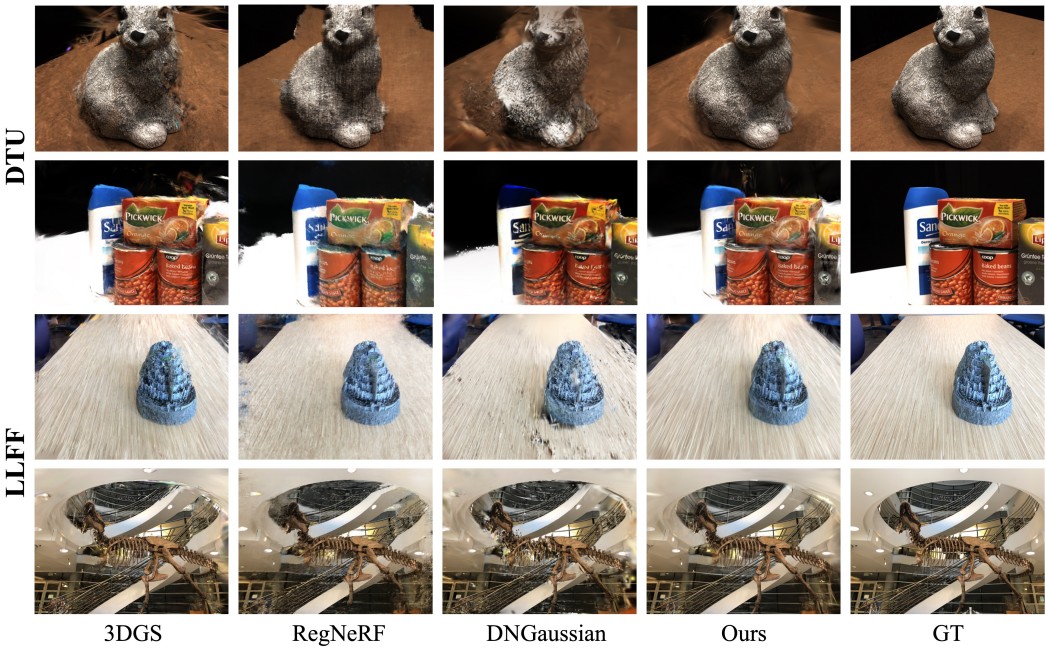

| 3DGS | RegNeRF | DNGaussian | Ours | GT |

Figure 2: **Qualitative comparison on DTU [12] and LLFF [19] datasets.** The results show that RegNeRF tends to produce blurred outcomes. 3DGS and DNGaussian introduce artifacts in the novel view. In contrast, our method generates better qualitative results.

### 4.1 Setup

**Datasets.** Our approach is evaluated on commonly used real-word [12, 19] and synthetic [20] datasets. Following previous methods, 15 scenes are selected on the DTU [12] dataset in our experiments, in which the background is removed during evaluation. LLFF [19] dataset is composed of 8 real-world scenes. Following FreeNeRF [40], we choose the 8th image for inference and then uniformly sample sparse views from the remaining images as the training set. For both datasets, the training set consists of 3 views. We use $\frac{1}{4}$ resolution images on DTU and $\frac{1}{8}$ resolution on LLFF. Synthetic Blender dataset [20] contains 8 scenes. We use the same setup as in [11].

**Evaluation Metrics.** We use PSNR [35], SSIM, and LPIPS [45] to measure rendering performance. For LPIPS, a lower value denotes better performance, while a higher value is preferable for PSNR and SSIM. Our method is compared to NeRF-based and concurrent 3DGS-based methods, including NeRF[20], SRF [5], PixelNeRF [43], MVSNeRF [3], Mip-NeRF [1], DietNeRF [11], RegNeRF [22], FreeNeRF [40], SparseNeRF [34], 3DGS [13], DNGaussian [15], FSGS [46], DRGS[6], and SparseGS [38].

**Implementation details.** Our method is trained for 10000 iterations in total on all the datasets. We use the same schedule by spending 2000 iterations on the pre-training stage, 7500 on the intermediate stage, and 500 on the tuning stage. In the pre-training stage and tuning stages, loss weights $\{\lambda, \chi, \zeta\}$ are set to $\{1.0, 0.001, 0.001\}$ respectively. In the intermediate stage, geometry, color, and semantic loss weights $\alpha, \beta, \gamma$ are set to 0.5, 0.05, and 0.001. Consistency and pre-training loss weights $\kappa, \eta$ are set to 1.0 and 0.05. $\delta$ in the locality preserving regularization is set to 2.0. The gradient threshold $\theta_{grad}$ is 0.1. $\theta$ in the agreement masking is 10.0. The scaling factor for the locality preserving loss $\delta$ is set to 0.2. We use a pre-trained VGG16 to extract semantic features. The experiments are run on one NVIDIA A6000 GPU.

### 4.2 Quantitative Results

We report our rendering performance on commonly used real-world datasets in Tab. 1. We compare our method against NeRF-based methods and concurrent Gaussian-based approaches (marked with an asterisk *). Our method performs on par or better than most of the presented approaches significantly improving over the baseline. On the DTU dataset, FreeNeRF [40] performs slightly better since it

Table 1: **Quantitative evaluation on DTU[12] and LLFF[19].** We use 3 training views across all the datasets. Concurrent works are marked with an asterisk*.

| | Setting | DTU | | | LLFF | | |
|---|---|---|---|---|---|---|---|
| | | PSNR ↑ | SSIM ↑ | LPIPS ↓ | PSNR ↑ | SSIM ↑ | LPIPS ↓ |
| SRF[5] | | 15.32 | 0.671 | 0.304 | 12.34 | 0.250 | 0.591 |
| PixelNeRF[43] | Trained on DTU | 16.82 | 0.695 | 0.270 | 7.93 | 0.272 | 0.682 |
| MVSNeRF[3] | | 18.63 | 0.769 | 0.197 | 17.25 | 0.557 | 0.356 |
| Mip-NeRF[1] | | 8.68 | 0.571 | 0.353 | 14.62 | 0.351 | 0.495 |
| DietNeRF[11] | | 11.85 | 0.633 | 0.314 | 14.94 | 0.370 | 0.496 |
| RegNeRF[22] | Optimized per Scene | 18.89 | 0.745 | 0.190 | 19.08 | 0.587 | 0.336 |
| FreeNeRF[40] | | **19.92** | 0.787 | 0.182 | 19.63 | 0.612 | 0.308 |
| SparseNeRF[34] | | 19.55 | 0.769 | 0.201 | 19.86 | 0.624 | 0.328 |
| 3DGS[13] | | 16.94 | 0.816 | 0.152 | 19.48 | 0.664 | 0.220 |
| DRGS[6]* | | - | - | - | 17.17 | 0.497 | 0.337 |
| SparseGS[38]* | | 18.89 | 0.702 | 0.229 | - | - | - |
| DNGaussian[15]* | Optimized per Scene | 18.23 | 0.780 | 0.184 | 18.86 | 0.600 | 0.294 |
| FSGS[46]* | | - | - | - | 20.43 | 0.682 | 0.248 |
| Ours (Rand. Init.) | | 19.13 | 0.792 | 0.186 | 18.96 | 0.585 | 0.307 |
| Ours | | 19.74 | **0.861** | **0.127** | **20.54** | **0.693** | **0.214** |

Table 2: **Qualitative evaluation.** We compare rendering with (*w*) and without (*w/o*) locality regularization, clearly indicating its effectiveness.

Table 3: **Quantitative evaluation on Blender[20] dataset. FewViewGS** shows superior performance using 8 training views.

w/o locality reg.          w locality reg.          GT

| Method | PSNR ↑ | SSIM ↑ | LPIPS ↓ |
|---|---|---|---|
| NeRF[20] | 14.934 | 0.687 | 0.318 |
| DietNeRF[11] | 23.147 | 0.866 | 0.109 |
| FreeNeRF[40] | 24.259 | 0.883 | 0.098 |
| SparseNeRF[34] | 22.410 | 0.861 | 0.119 |
| 3DGS[13] | 22.226 | 0.858 | 0.114 |
| DNGaussian[15]* | 24.305 | **0.886** | **0.088** |
| Ours | **25.550** | **0.886** | 0.092 |

hallucinates over the missing regions, while Gaussian Splatting can't due to its explicit geometry encoding. Our method shows SoTA results on LLFF datasets. Interestingly, our method shows compelling results even with random Gaussian initialization. Finally, our method shows state-of-the-art results on the synthetic Blender dataset in Tab. 3.

### 4.3 Ablation Study

We ablate over our key design choices presented in the paper on the DTU dataset with 3 training views in Tab. 4, in which random initialization is used for 3D Gaussians. In the ablations, the effects of each contribution are examined in isolation.

**Locality Preserving Regularization.** The locality preservation loss improves rendering when added standalone to the vanilla Gaussian Splatting (row *ii*) and when added to the consistency losses in a multi-stage training setup (row *xiv*).

**Novel View Consistency.** We ablate over novel view consistency losses and show our results in rows *iii* to *ix*. First, it is shown in rows *iii* to *v* that geometry, appearance, and semantic losses contribute to the higher quality rendering when added separately. In particular, geometry and color consistency remarkably improve the performance, since they directly influence the attributes of 3D Gaussians and are highly related to the view-synthesis task. In rows *iii, iv, v* it is shown that all three consistency losses lead to even better results. Experiments are conducted on different large-scale feature extraction backbones for the semantic loss, such as DINOv2 [23], CLIP [27], and VGG [30] in *vii, viii, ix*. The results demonstrate that DINOv2 and CLIP perform worse than VGG. This occurs because DINO and CLIP significantly down-sample features early in their encoders, leading to missing details and

Table 4: **Ablation study on DTU [12] dataset under 3-view setting.**

|  | Method | PSNR ↑ | SSIM ↑ | LPIPS ↓ |
|---|---|---|---|---|
| i | 3DGS [13] | 15.04 | 0.676 | 0.246 |
| ii | $L_{locality}$ | 15.64 | 0.683 | 0.243 |
| iii | $L_{geom}$ | 18.17 | 0.736 | 0.198 |
| iv | $L_{color}$ | 18.50 | 0.773 | 0.192 |
| v | $L_{sem}$ | 15.52 | 0.691 | 0.235 |
| vi | $L_{geom} + L_{color}$ | 18.87 | 0.791 | 0.186 |
| vii | $L_{geom} + L_{color} + L_{sem}$ (DINOv2 [23]) | 18.71 | 0.787 | 0.188 |
| viii | $L_{geom} + L_{color} + L_{sem}$ (CLIP [27]) | 17.86 | 0.762 | 0.200 |
| ix | $L_{geom} + L_{color} + L_{sem}$ (VGG [30]) | 18.95 | 0.787 | 0.186 |
| x | $L_{geom} + L_{color} + L_{sem}$ (w/o min.) | 18.28 | 0.784 | 0.194 |
| xi | $L_{geom} + L_{color} + L_{sem}$ (single-stage) | 18.15 | 0.783 | 0.186 |
| xii | $L_{geom} + L_{color} + L_{sem}$ (w/o matching) | 16.05 | 0.717 | 0.224 |
| xiii | $L_{geom} + L_{color} + L_{sem}$ (SIFT [17]) | 16.04 | 0.704 | 0.238 |
| xiv | $L_{geom} + L_{color} + L_{sem} + L_{locality}$ (ours) | 19.13 | 0.792 | 0.186 |

blurring, especially in the boundary regions, as illustrated in Fig. 6. In contrast, our method utilizes the low-level features of VGG16 for the semantics constraint, balancing the positive effect from local semantic features and missing details.

**Minimum Loss in Consistency Losses.** We demonstrate in the row $x$ the effect of the $min$ operation in Eq. (9) - Eq. (11). $min$ softens the constraints imposed on the 3D Gaussians by reducing the influence of errors in feature matching and rendered depth.

**Multi-stage Training.** Row *xi* reports results with single-stage training, in which training views and novel views are utilized during optimization. Compared to the multi-stage training in row *ix*, the single-stage regime performs worse, since the Gaussian parameters tend to overfit to the training views. Moreover, without proper scene initialization, the depth maps for the training views become contaminated by the novel views at the start of the training procedure.

**Feature Matching.** Experiments are also conducted using only a single known view for the consistency losses in row *xii*. We utilize a single training view, sample a random novel view in its frustum view, and reproject the points to it for additional supervision. There's a drastic difference in rendering performance compared to the variant where feature matching is not used in *xii*. This happens because the rendered depth in a training view may contain many errors that mislead the novel view synthesis. Feature matching algorithm plays an important role by choosing points appearing in several views for the warping, providing the ability to compute the agreement mask for filtering out unreliable color, depth, and features. RoMa [9] is particularly effective since it provides many correspondences between the images that are quite far from each other. Experiments are conducted with a classical feature matching algorithm [17] in *xiii*, but it fails due to a limited number of matches (see Fig. 7).

## 5   Conclusion

We introduced **FewViewGS**, a few-shot novel view synthesis system based on 3D Gaussian Splatting as the scene representation that enables accurate renderings using only a handful of training images. We proposed an effective multi-stage training scheme, novel view consistency constraints, and regularization losses. Compared to previous state-of-the-art sparse novel view synthesis systems, we achieve high-quality rendering without relying on complex priors like depth estimation or diffusion models. We demonstrated that **FewViewGS** yields compelling results in rendering on both real-world and synthetic datasets.

**Limitations.** Our method may struggle to render texture-rich regions accurately, particularly when novel views diverge significantly from the input views. Additionally, as indicated by our ablation study, utilizing a more precise feature-matching network to create dense matched pairs would be beneficial, as this can offer more robust supervision for the novel views.

## Acknowledgement

Ruihong Yin is financially supported by China Scholarship Council and University of Amsterdam. Vladimir Yugay and Yue Li are financially supported by TomTom, the University of Amsterdam and the allowance of Top consortia for Knowledge and Innovation (TKIs) from the Netherlands Ministry of Economic Affairs and Climate Policy.

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

# A  Appendix

## A.1  Additional Quantitative Results

**Comparison with SOTA methods under 6-view and 9-view settings.** The results for 6-view and 9-view settings are given in  Tab. 5. Our method shows improved performance over the baseline 3DGS with both SFM and random initialization. Notably, with SFM initialization, our method surpasses NeRF-based methods on both datasets.

**Comparison with SOTA methods on training and inference time.** Tab. 11 displays the training time on a single NVIDIA A6000 GPU under 3-view setting on DTU. Our method demonstrates comparable training time and is faster than SparseGS [38] (which uses a pre-trained diffusion model) and FSGS [46] (which relies on a pre-trained depth network). For inference time, since all methods utilize the same rendering process as 3DGS, their inference time is similar across these methods, at around 300 FPS.

**Ablation study for hyperparameters.** We provide evaluations for hyperparameters and the multi-stage settings in  Tab. 6 -  Tab. 10 on DTU with 3 training views. (1) $\alpha$ in  Eq. (12): we set it to a low weight due to the lack of ground truth for the depth. It can be seen that, in  Tab. 7, larger (e.g. 0.2) gets poorer results, primarily due to noise in predicted depth values. (2) $\beta$ in  Eq. (12): due to the importance of color supervision for the novel view synthesis, a large weight is utilized in the experiment. In  Tab. 6, a value of 0.5 achieves the best performance. (3) $\gamma$ in  Eq. (12): The results in  Tab. 9 demonstrate that our method is robust to different $\gamma$ values, with $\gamma = 0.001$ being optimal. (4) $\eta$ in  Eq. (15): The pre-training loss in the intermediate stage is employed to ensure sufficient supervision for novel views. In  Tab. 8, a lower $\eta$ (0.01) does not solve this very well, while higher $\eta$ (0.1 and 0.5) fails to prevent overfitting to the training views. (5) Multi-stage settings: we evaluate the influence of iterations for each stage in our 3-stage training in  Tab. 10. The last two rows show that a fewer/more iteration in the first stage leads to worse results, due to underfitting/overfitting to training views. The 1st-5th rows demonstrate that the fine-tuning stage enhances performance and achieves optimal results with 500 iterations.

Table 5: **Quantitative evaluation on LLFF and DTU under 6-view and 9-view settings.**

| | Setting | Init. | 6-view (LLFF) | | | 9-view (LLFF) | | |
|---|---|---|---|---|---|---|---|---|
| | | | PSNR ↑ | SSIM ↑ | LPIPS ↓ | PSNR ↑ | SSIM ↑ | LPIPS ↓ |
| SRF | | - | 13.10 | 0.293 | 0.594 | 13.00 | 0.297 | 0.605 |
| PixelNeRF | Trained on DTU | - | 8.74 | 0.280 | 0.676 | 8.61 | 0.274 | 0.665 |
| MVSNeRF | | - | 19.79 | 0.656 | 0.269 | 20.47 | 0.689 | 0.242 |
| Mip-NeRF | | - | 20.87 | 0.692 | 0.255 | 24.26 | 0.805 | 0.172 |
| DietNeRF | Optimized per Scene | - | 21.75 | 0.717 | 0.248 | 24.28 | 0.801 | 0.183 |
| RegNeRF | | - | 23.10 | 0.760 | 0.206 | 24.86 | 0.820 | 0.161 |
| FreeNeRF | | - | 23.73 | 0.779 | 0.195 | 25.13 | 0.827 | 0.160 |
| 3DGS | Optimized per Scene | Random | 19.81 | 0.638 | 0.245 | 22.32 | 0.753 | 0.169 |
| Ours | | Random | 21.33 | 0.688 | 0.220 | 23.09 | 0.769 | 0.164 |
| 3DGS | Optimized per Scene | SFM | 23.65 | 0.807 | **0.123** | 25.26 | 0.852 | 0.096 |
| Ours | | SFM | **24.35** | **0.826** | 0.126 | **25.90** | **0.868** | **0.095** |
| | Setting | Init. | 6-view (DTU) | | | 9-view (DTU) | | |
| | | | PSNR ↑ | SSIM ↑ | LPIPS ↓ | PSNR ↑ | SSIM ↑ | LPIPS ↓ |
| SRF | | - | 17.77 | 0.616 | 0.401 | 18.56 | 0.652 | 0.359 |
| PixelNeRF | Trained on DTU | - | 21.02 | 0.684 | 0.340 | 22.23 | 0.714 | 0.323 |
| MVSNeRF | | - | 18.26 | 0.695 | 0.321 | 20.32 | 0.735 | 0.280 |
| Mip-NeRF | | - | 14.33 | 0.568 | 0.394 | 20.71 | 0.799 | 0.209 |
| DietNeRF | Optimized per Scene | - | 18.70 | 0.668 | 0.336 | 22.16 | 0.740 | 0.277 |
| RegNeRF | | - | 19.10 | 0.757 | 0.233 | 22.30 | 0.823 | 0.184 |
| FreeNeRF | | - | 22.39 | 0.779 | 0.240 | 24.20 | 0.833 | 0.187 |
| 3DGS | Optimized per Scene | Random | 20.46 | 0.824 | 0.145 | 24.75 | 0.914 | 0.076 |
| Ours | | Random | 23.51 | 0.891 | 0.123 | 25.75 | 0.925 | 0.101 |
| 3DGS | Optimized per Scene | SFM | 23.63 | 0.912 | 0.074 | 26.53 | 0.946 | 0.047 |
| Ours | | SFM | **24.33** | **0.920** | **0.069** | **27.31** | **0.953** | **0.041** |

Table 6: **Ablation study for the hyperparameter $\beta$ in Eq. (12).**

| $\beta$ | PSNR ↑ | SSIM ↑ | LPIPS ↓ |
|---|---|---|---|
| 0.1 | 18.68 | 0.785 | **0.186** |
| 0.5 | **19.13** | **0.792** | **0.186** |
| 1.0 | 18.70 | 0.784 | 0.191 |

Table 7: **Ablation study for the hyperparameter $\alpha$ in Eq. (12).**

| $\alpha$ | PSNR ↑ | SSIM ↑ | LPIPS ↓ |
|---|---|---|---|
| 0.01 | 18.67 | 0.778 | 0.192 |
| 0.05 | **19.13** | **0.792** | **0.186** |
| 0.1 | 18.92 | 0.790 | 0.190 |
| 0.2 | 18.50 | 0.786 | 0.196 |

Table 8: **Ablation study for the hyperparameter $\eta$ in Eq. (15).**

| $\eta$ | PSNR ↑ | SSIM ↑ | LPIPS ↓ |
|---|---|---|---|
| 0.01 | 18.89 | 0.781 | 0.201 |
| 0.05 | **19.13** | **0.792** | 0.186 |
| 0.1 | 18.58 | 0.785 | 0.183 |
| 0.5 | 18.20 | 0.772 | **0.179** |

Table 9: **Ablation study for the hyperparameter $\gamma$ in Eq. (12).**

| $\gamma$ | PSNR ↑ | SSIM ↑ | LPIPS ↓ |
|---|---|---|---|
| 0.001 | **19.13** | **0.792** | 0.186 |
| 0.01 | 18.84 | 0.785 | 0.186 |
| 0.1 | 18.85 | 0.787 | 0.187 |
| 0.5 | 19.03 | **0.792** | **0.184** |
| 1.0 | 18.88 | 0.789 | 0.188 |

Table 10: **Ablation study for the multi-stage settings.**

| 1st | 2nd | 3rd | PSNR ↑ | SSIM ↑ | LPIPS ↓ |
|---|---|---|---|---|---|
| 2000 | 8000 | 0 | 18.49 | 0.782 | 0.195 |
| 2000 | 7500 | 0 | 18.37 | 0.782 | 0.198 |
| 2000 | 7200 | 800 | 18.85 | 0.787 | 0.187 |
| 2000 | 7500 | 500 | **19.13** | **0.792** | **0.186** |
| 2000 | 7800 | 200 | 18.71 | 0.786 | 0.191 |
| 1000 | 8500 | 500 | 18.72 | 0.772 | 0.228 |
| 3000 | 6500 | 500 | 18.76 | 0.782 | 0.189 |

Table 11: **Comparison of training time.**

| Method | Time ↓ |
|---|---|
| 3DGS | 1.17 min |
| SparseGS | 51.78 min |
| DNGaussian | 2.65 min |
| FSGS | 10.90 min |
| Ours | 5.82 min |

## A.2 Additional Qualitative Results

**Visualization of unseen regions.** In Fig. 3, we compare the results of unseen regions between our method and FSGS [46]. The visualization shows that our method produces fewer artifacts and outperforms FSGS (which relies on a pre-trained depth network). This improvement is due to our multi-view consistency constraints, which offer accurate supervision for seen regions, while the proposed locality-preserving regularization maintains local smoothness and reduces artifacts.

**Visualization of predicted depth.** Fig. 4 presents the generated depth by 3DGS [13] and our method. Our method produces clearer and more accurate depth estimations than the baseline 3DGS, demonstrating its effectiveness.

**Ablation study on multi-view constraints.** In Fig. 5, qualitative results are shown of our proposed multi-view geometry, color, and semantic constraints. The results indicate that each design choice contributes to improved rendering quality.

**Ablation study on various networks for semantic feature extraction.** Fig. 6 compares our results (VGG16) with DINOv2 and CLIP. The results indicate that DINOv2 and CLIP introduce substantial noise, particularly in boundary regions, and detail preservation is hindered by feature down-sampling in both models.

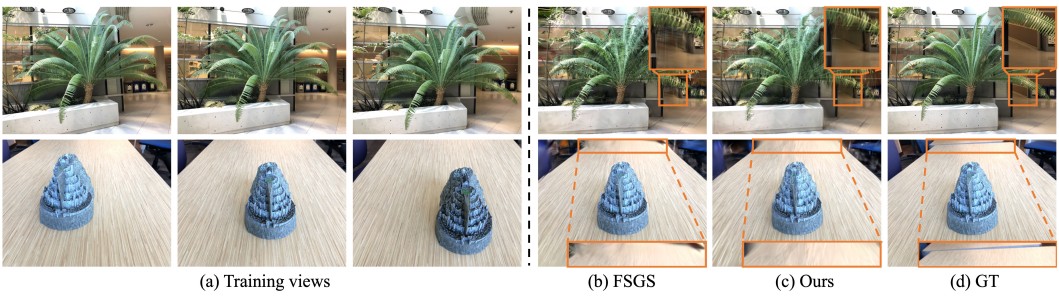

    (a) Training views        (b) FSGS    (c) Ours    (d) GT

Figure 3: **Visualizations for unseen regions.** The orange regions are not observed in the training views. Compared to FSGS [46], our method generates better results and fewer artifacts.

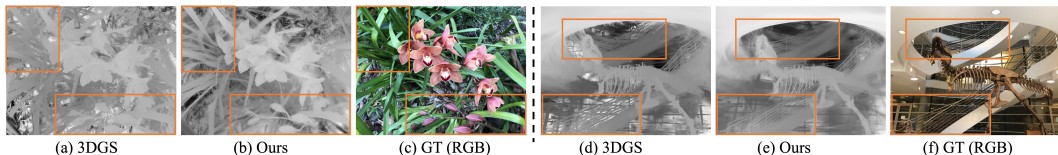

(a) 3DGS    (b) Ours    (c) GT (RGB)    (d) 3DGS    (e) Ours    (f) GT (RGB)

Figure 4: **Visualizations for predicted depth.** Compared to the baseline 3DGS [13], our method yields more accurate depth values.

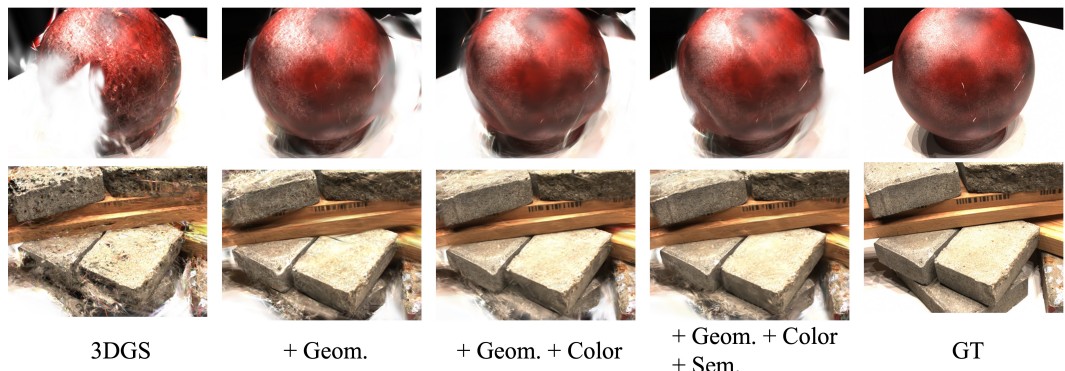

3DGS    + Geom.    + Geom. + Color    + Geom. + Color + Sem.    GT

Figure 5: Comparison of the results with our proposed multi-view alignment.

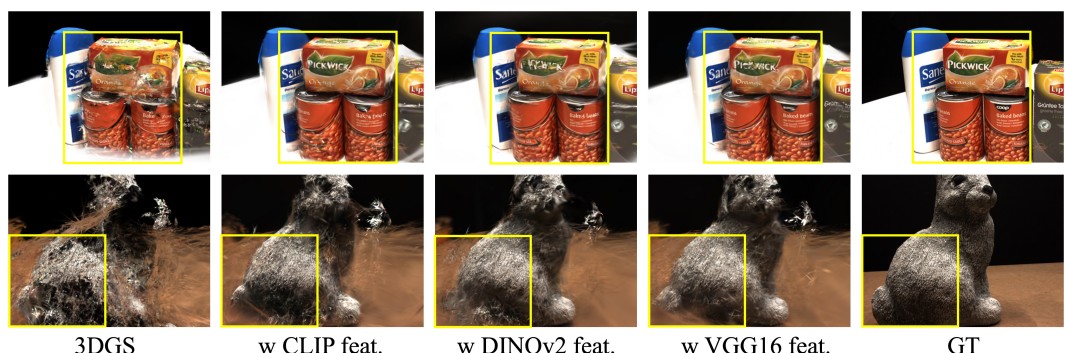

3DGS    w CLIP feat.    w DINOv2 feat.    w VGG16 feat.    GT

Figure 6: Comparison of the results with different feature networks.

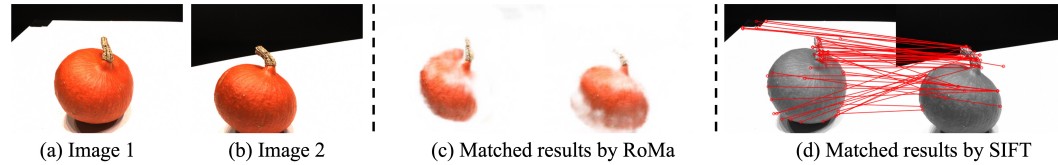

(a) Image 1    (b) Image 2    (c) Matched results by RoMa    (d) Matched results by SIFT

Figure 7: Comparison of the matched results with different feature-matching algorithms. Roma[9] and SIFT[17] implement feature matching between the *Image1* and *Image2*.

**Ablation study on feature matching.** In Fig. 8, the results with and without feature matching are presented. It is shown that the model without feature matching generates false surfaces due to noise in the multi-view projection.

In Fig. 7, we show the difference between matching results generated by RoMa and SIFT. RoMa produces denser and more accurate matches than SIFT, which significantly enhances the model's performance when using RoMa.

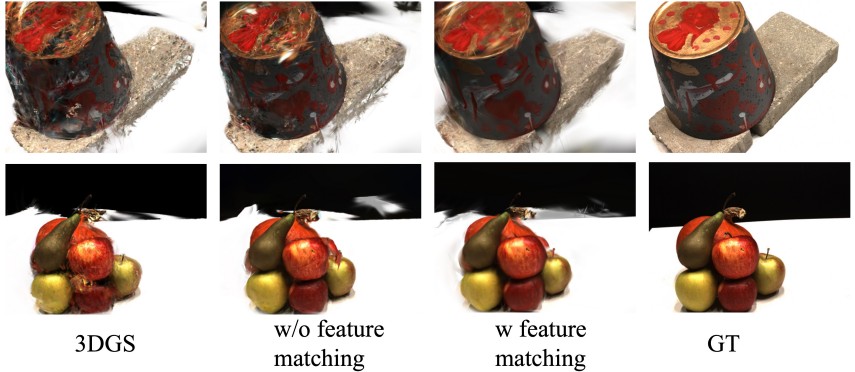

|        | w/o feature | w feature |     |
|--------|-------------|-----------|-----|
| 3DGS   | matching    | matching  | GT  |

Figure 8: Comparison of the results without and with feature matching results.

**Ablation study on multi-stage training.** The comparison between single-stage and multi-stage training is demonstrated in Fig. 9. Single-stage training results in blurry views due to inaccurate depth rendering in the early training stages, leading to errors in multi-view projection.

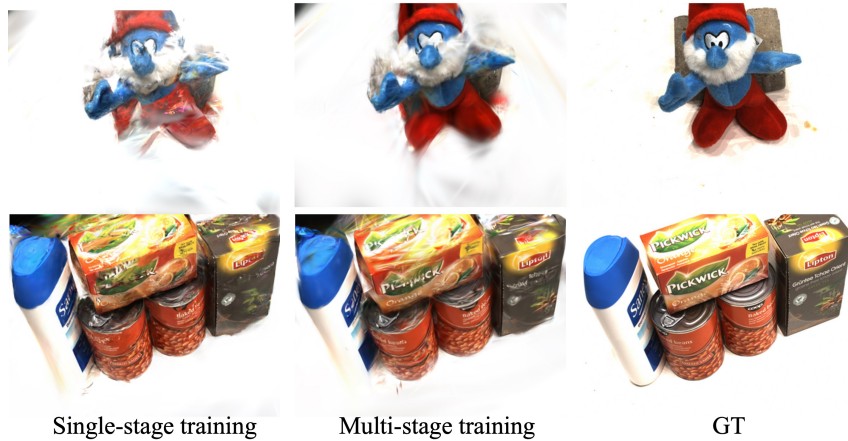

Single-stage training          Multi-stage training          GT

Figure 9: Comparison between single-stage and multi-stage training with our multi-view consistency constraints.

