# OpenReview forum: "FewViewGS: Gaussian Splatting with Few View Matching and Multi-stage Training"
_NeurIPS.cc/2024/Conference — NeurIPS 2024 poster_

### Official Review · Reviewer_oLUf · 2024-06-25

**Soundness:** 2
**Presentation:** 2
**Contribution:** 2
**Rating:** 3
**Confidence:** 5

**Summary:**

This paper presents FewViewGS to regularize sparse view 3DGS from unseen viewpoints without relying on pre-trained depth estimators. The main contribution is to reproject the pixels to an unseen view and calculate the losses at corresponding pixels found by image matching. Experiments on LLFF, DTU, and Blender datasets demonstrate the proposed method can achieve competitive or superior performance compared to existing SOTA methods.

**Strengths:**

- The method can exceed existing baselines without relying on extra depth estimators.

- Results are good in visualization.

**Weaknesses:**

- The main contribution of this paper shares the same insight with SPARF [1], which also uses image matching to add supervision on unseen views. The main difference is that SPARF uses depth, while this work uses depth, color and semantic features, for which the novelty in this is limited. And also the multi-stage training is just a common composition of " initialization + regular training + refinement".

- The proposed equations have too many manual factors and artificial designs that lack mathematical theory (e.g. Eq (7, 8,13)). These all seem to be engineering attempts but not technical contributions.

- Experiment is insufficient and unclear. 1) Although a  (Rand. Init.) is mentioned at one item in Table 1, I can not find any description of the used initialization for other settings, which may have a very huge influence on the performance. 2) And serving as a geometry reconstruction method, some geometry visualizations like depth are wished to report. 3) As the reprojection operation and semantic feature extraction may cost a lot of time,  I'm wondering if the training time would significantly increase.

- Performance improvements are somewhat limited compared to current methods. And although this work can be free from some pre-trained models, it has introduced other dependencies like VGG features and RoMa matching.

- The paper has some errors, for example: 1) Eq (6) is unclear on the part (C, Z), which is a 2D->3D back projection I guess.  2) The citation information of SPARF [1] (numbered as [34] in the paper) is wrong.

[1] Truong, Prune, et al. "Sparf: Neural radiance fields from sparse and noisy poses." Proceedings of the IEEE/CVF Conference on Computer Vision and Pattern Recognition. 2023.

**Questions:**

Please clarify the weaknesses, and make additional explanations of the unclear parts.

**Limitations:**

The authors have not discussed the limitations. This work seems to heavily rely on some manual hyperparameters, and some pre-trained models like VGG features and RoMa matching, which may limit the applicability to more real-world scenes.

---

> ### Author Rebuttal · Authors · 2024-08-07
>
> We thank the reviewer for the insightful review. Below, we address the main concerns raised in this review.
>
> ***Q: The main contribution of this paper shares the same insight with SPARF.***
>
> ***A:*** Although both our method and SPARF target solving the overfitting issue, there are key differences that contribute to our novelty and effectiveness: (1) **Feature matching**: Our method samples novel views and projects matched pixels to the novel view for supervision. However, SPARF only adds additional supervision on matched pixels for training views, without involving any novel views. (2) **Supervision of novel views**: Our method projects two training views to generate supervision for novel views, improving the reliability of supervision. This contrasts with SPARF's projection including one training view, which can suffer from noises in depth estimation. Our experiments in Table 4ix and 4xii show that our method with two-view projection for novel view supervision notably outperforms the single-view projection. We appreciate the reviewers' suggestions and will add these comparisons to our revised version.
>
> ***Q: The multi-stage training is just a common composition of " initialization + regular training + refinement".***
>
> ***A:*** While we follow a common training composition, our 3-stage method aims to address overfitting to sparse training views and improve supervision reliability in novel views. Particularly, our approach introduces a novel method to integrate additional view information during the intermediate training stage. We apply feature matching to identify corresponding pixels, which are then projected onto novel views to generate pseudo labels. This approach further uses knowledge from training views, ensures multi-view consistency, enhances reliability during projection, and solves overfitting. Besides, our proposed locality-preserving regularization maintains the local smoothness of 3D Gaussians, enhancing the effectiveness of our method.
>
> ***Q: The proposed equations have too many manual factors and artificial designs that lack mathematical theory (e.g. Eq (7, 8,13)).***
>
> ***A:***. (1) **Hyperparameters**: We apologize for missing the relevant ablation study. Table S2 - Table S6 of the uploaded file provide additional results, showing that our method performs consistently well across different settings. We will include these results in our revised appendix. (2) **Mathematical theory**: The proposed equations are designed based on the principles of multi-view projection (Eqs 7 and 8) and the theory of local smoothness in 3D space (Eq 13). **Eq 7** is based on the principle that matched 2D pixels should map to the same 3D point. Thus, it measures the distance between 3D points to filter out noisy projections, which can solve depth inaccuracies in multi-view projection. **Eq 8** tackles the problem of blurry results in texture-rich regions with large gradients during projection, which are sensitive to projection errors. **Eq 13**, incorporating local constraints using the KNN algorithm, is based on spatially local smoothness.
>
> ***Q: 1) I can not find any description of the used initialization for other settings. 2) Some geometry visualizations like depth are wished to report. 3) I'm wondering if the training time would significantly increase.***
>
> ***A:*** We appreciate the reviewer's suggestions on experiments. We will incorporate additional details and results in our revised version. (1) **Initialization**: In Table 1, our method with random and SFM initialization is listed in the last two rows. In the ablation study, the random initialization was adopted. (2) **Geometry visualization**: In Figure S2 of the uploaded file, our method generates clearer and more accurate depths compared to the baseline 3DGS, demonstrating its effectiveness. (3) **Training time**: Table S7 of the uploaded file compares the training time of 3DGS-based methods. Our method has comparable training time and is faster than SparseGS (which uses a pre-trained diffusion model) and FSGS (which relies on a pre-trained depth network).
>
> ***Q: Performance improvements are somewhat limited compared to current methods. And although this work can be free from some pre-trained models, it has introduced other dependencies like VGG features and RoMa matching.***
>
> ***A:*** (1) **Performance:** In Table 1, our method surpasses 3DGS-based methods over all metrics on both DTU and LLFF datasets. Specifically, on DTU, compared to SparseGS, our method improves PSNR by 4.5\% and SSIM by 22.6\%. These results demonstrate the effectiveness of our method. We will integrate additional relevant descriptions to our manuscript. (2) **Pre-trained networks:** Previous methods, such as FSGS and SparseGS, use pre-trained depth/diffusion models to create pseudo labels for novel views, which may be noisy. In contrast, our method takes a different approach by projecting the color data (ground truth) from training views to novel views to yield pseudo labels, which is more accurate than the labels produced by other networks. Specifically, the accuracy of this projection is maintained using RoMa matching, a technique that finds correspondence between training views. In addition to color supervision, our method also employs the VGG network to provide further semantic supervision for novel views. These techniques help maintain multi-view consistency and prevent overfitting.
>
> ***Q: 1) Eq (6) is unclear on the part (C, Z), which is a 2D->3D back projection I guess. 2) The citation information of SPARF is wrong.***
>
> ***A:*** We apologize for the confusion caused. In Eq 6, as explained in Lines 159-164, *C* denotes 2D coordinates of pixels, while *Z* refers to their depth values. The process in Eq 6 involves a 2D-3D projection (i.e. $\pi^{-1}(C_i, Z_i)$) and a 3D-2D projection (i.e. $\pi(P_k \cdot)$). We will make it more clear in the next version. (2) Thank you for pointing that out. We will check the citations and ensure they are correct.

---

> > ### Comment · Reviewer_oLUf · 2024-08-08
> >
> > Thanks for the reply from the authors. After reading the rebuttal, this work still has the following problems.
> >
> > 1. Lack of Novelty. Summarized by the authors in the paper, this work has three main contributions: an effective multi-stage training scheme, novel view consistency constraints, and a color smooth regularization loss. First, as in the review, the multi-stage training is just a common composition of " initialization + regular training + refinement", with no independent novelty on this point. And the color smooth regularization loss is just a trick to add local smoothness to reduce some artifacts, which only improved PSNR but not SSIM and LPIPS, demonstrating its limited effect on the overall quality. If these are declared to be the most important novelties in the paper, I can only believe there are no other valuable things to show.
> >
> >     And for the novel view consistency constraints, it is still the incremental technique of SPARF even according to the rebuttal. The contribution lies only in some loss designs with no interesting insight, moreover, has too many manual hyperparameters in eq(7,9,10,11) and replaceable mapping functions like $exp()$ in eq(8, 13).  Besides, L_sem is less effect according to Table 4 (vi, ix, xiv) and Figure 4. Especially, some details became worse after adding L_sem in Figure 4. It is also sensitive to the type of feature. I do not think these novelties can match the bar at NeurIPS.
> >
> > 2) Details of "SFM initialization" is still not clear enough. Please explain how many views are involved in the SfM process to get the camera poses and point cloud, how to align the excluded test views into the camera coordinate system, and whether is there any post-processing of the point cloud.  As far as I know, original 3DGS can not recover such precise scenes shown in Figure S2 when just use 3-view SfM point cloud as initialization, and COLMAP will fail in some DTU scenes when using only three training views. I have doubts about these.
> >
> > 3) The authors claimed introducing depth or diffusion models will "make training time longer and highly dependent on the quality of the pre-trained models" (lines 40-41), which is the stem of this work. However, why introducing pre-trained RoMa and VGG can escape from these weaknesses? The logic is weird.
> >
> > I'll temporarily keep my rating.

---

> ### Author Response · Authors · 2024-08-10
> **Reply to Reviewer oLUf**
>
> We sincerely appreciate your time and efforts in reviewing our responses. We hope that our responses below adequately address your concerns.
>
> ***Q1: Clarification of our novelty***
>
> ***A1:*** In this response, we would like to clarify our novelty and design details.
> **(1) Novel view consistency constraints:** Our method offers a new solution to address overfitting and ensure multi-view consistency in sparse view synthesis by using correspondence priors to generate labels for novel views. Since accurate depth prediction is crucial for reliable view projection, our method projects matched pairs from training views to the novel view and filters out pairs with significant depth discrepancies in the novel view (see Eq 7). This design enhances label reliability over SPARF's single-view projection. The table below shows the results of SPARF's multi-view correspondence loss (MV-Corr) and depth consistency loss (DCons) on 3DGS, which is much worse than our method (PSNR 19.13, SSIM 0.792, LPIPS 0.186). Furthermore, DCons in SPARF shows limited improvement, see Row i and Row ii. In contrast, our depth supervision for novel views in Table 4iii (PSNR 18.17, SSIM 0.736, and LPIPS 0.198) greatly improves the performance of 3DGS.
> || Method | PSNR | SSIM |LPIPS |
> | --- | --- | --- | ---| --- |
> | i| 3DGS | 15.04  | 0.676 |   0.246 |
> | ii | + DCons | 15.17 | 0.699  |   0.232 |
> | iii| + MV-Corr | 16.25 | 0.722 |  0.219 |
> | iv| + MV-Corr + DCons | 16.37 |  0.729| 0.214 |
>
> **(2) Multi-stage training:** Common multi-stage training typically involves an additional teacher network or large dataset. In contrast, our approach tailors the multi-stage training to sparse view synthesis, eliminating the need for extra networks or datasets. It alternates between training and novel views, expanding the view information and solving overfitting. As shown in Table 4ix and 4xi, this multi-stage approach improves the PSNR by 4.4\%. **(3) Locality preserving regularization:** Our locality regularization explicitly constrains the color attributes of 3D Gaussians to preserve local smoothness in 3D space. Table 2 demonstrates reduced artifacts and improved accuracy, and Table 4i and 4ii highlight gains in all metrics. **(4) Loss design:** Our proposed loss functions integrate additional constraints directly, which is intuitive and effective to address issues in sparse view synthesis, without increasing inference time. **(5) $L_{sem}$:** Our work introduces the semantic constraint to integrate local constraints and reduce artifacts. The comparisons between Table 4i and 4v show a 3.2\% increase in PSNR and a 4.5\% reduction in LPIPS. Indeed, as discussed in Lines 272-277, the local constraint needs to balance the benefits of local semantics with the risk of missing details. **(6) Other details:** We apply consistent values for $\theta$ and $\theta_{grad}$ in Eqs (9, 10, 11), to avoid too many human factors. Fine-tuning these parameters and exploring better mapping functions could improve results.
>
> ***Q2: Details of SFM initialization***
>
> ***A2:*** The code for SFM process will be released, providing all the details. **(1) SfM initialization**: We follow the SfM process provided in FSGS ('tools/colmap\_llff.py' in FSGS code) to generate camera poses and 3D points, using only training views without any post-processing. The choice of training/test views is consistent with FSGS on LLFF and DNGaussian on DTU. Novel views are randomly interpolated between training views. **(2) Visualizations**: Figure S2 shows 3DGS results generated with SfM initialization, a 3-view setting, and size_threshold=None (as used in  FSGS and DNGaussian). We will release the model for further clarification. **(3) COLMAP in DTU**: Following FSGS, we use the dense reconstruction in COLMAP to generate 3D points, with only training views, which successfully yields point clouds on all scenes except 'scan110'. They are used to initialize 3D Gaussians. 'Scan110' is initialized randomly.
>
> ***Q3: Comparison with other works based on pre-trained depth/diffusion models.***
>
> ***A3:*** We try to clarify the confusion below. **(1) Training time:** We pre-compute and store VGG features and RoMa-matched pairs during data pre-processing, avoiding extra training time and redundant computations. However, pre-trained depth/diffusion models generate results online, making them more computationally expensive than pre-trained Roma and VGG models. **(2) Quality:** Networks using pre-trained depth/diffusion models generate labels for the novel view often ignore the priors in training views and lack multi-view consistency. In contrast, our method projects labels from training views to novel views, ensuring higher accuracy and multi-view consistency. Using a pre-trained depth network for novel views on 3DGS yielded PSNR 16.74, SSIM 0.734, LPIPS 0.214. Our method with depth supervision alone in Table 4iii (PSNR 18.17, SSIM 0.736, and LPIPS 0.198) outperforms this.

---

> > ### Comment · Reviewer_oLUf · 2024-08-11
> >
> > I'll respond to some important problems, but not mean that other parts are good:
> >
> > 1. For **novel view consistency constraints**: According to the response, the authors agree that their work lacks novelty, since the novelty claimed by the authors, eq (7), is just a simple filter mask supported by a manual hyperparameter, while the other parts in "novel view consistency constraints“ is strongly similar to SPARF.
> >
> >     on the part of the effect, the provided reproduction experiment of SPARF on 3DGS is not convincing for me, as there is not any explanation about why the performance gap happens. BTW, according to its paper, SPARF can achieve the performance of PSNR 21.01, SSIM 0.87, LPIPS 0.10 on NeRF backbone in the 3-view DTU setting, which can already fully validate the effect of their strategies. Considering the working principle of this work is extremely close to SPARF without significant innovation, it can be considered as an incremental work to transfer SPARF to 3DGS backbone.
> >
> >  2. For **semantic loss**: According to Table 4 vi and ix, the improvement is very marginal. What's worse, when applying an unsatisfactory type of feature, the performance will drop significantly. On the other hand, Despite that the authors try to prove the effect through "The comparisons between Table 4i and 4v show a 3.2% increase in PSNR and a 4.5% reduction in LPIPS", the analysis is partial. It only compares the situations with raw 3DGS. According to the comparisons between Table 4 vi, vii, viii and ix, its effect significantly shrinks to none after other constraints are applied, showing its redundancy.
> >
> >
> > 3. For the **so-called "SFM initialization"**, it shows that the authors lack the basic knowledge in multi-view geometry and the ability of discernment. Following the description and code process of FSGS, the author declared they use "the SfM process ...  to generate camera poses and 3D points, using only training views without any post-processing". However, first, there is a significant mistake in FSGS that it calls the process they used as SfM but actually is an MVS method. Second, this process does not estimate any camera poses. So, the initialization method the authors used is actually MVS, rather than SfM they announced in the paper and rebuttal. This problem shows that the authors do not even know what they are actually doing. Considering this problem is to be revealed after the reviewer's twice asks, I'm worrying about this work's quality.
> >
> > 4. For **introducing pre-trained models**: the authors did not directly reply to my questions. First, are RoMa and VGG pre-trained models? Then, can these pre-trained models escape from the problems that "make training time longer and highly dependent on the quality of the pre-trained models"?  In other words, is there not any extra time cost in order to make the proposed method as fast as raw 3DGS, especially when needing to use VGG to generate the semantic embedding for novel view k in eq (11) online? Or can it also work well when using models with poor pre-training quality? If not, how can the authors claim that their method is different from previous works in this part?

---

> > > ### Author Response · Authors · 2024-08-13
> > > **Reply to Reviewer oLUf**
> > >
> > > We sincerely appreciate your time and efforts in reviewing our responses. We hope that our responses below adequately address your concerns.
> > >
> > > ***Q1: Clarification of our novelty***
> > >
> > > ***A1:*** We strongly disagree with the reviewer's statement about novelty. This assertion is not only inaccurate but also a misrepresentation of our previous responses. Our method aims to address the overfitting and maintain the multi-view consistency, as detailed in Section 3.2. While SPARF also aims to mitigate overfitting, a challenge common to sparse view synthesis, the similarity with our method ends there.
> > >
> > > Our method utilizes the correspondence priors to guide multi-view projection, filtering out outliers and generating reliable pseudo-labels, which have been verified to be crucial. Subsequently, we compute the appearance/geometry/semantic losses with gradient weighting for the novel view. This not only broadens the available view information but also maintains multi-view consistency. Notably, during loss computation, we again utilize correspondence priors to select the minimal loss among each matched pair, which can reduce the effect of large noises. Table 4ix and 4x prove the effectiveness of the minimal operation.
> > >
> > > Our method diverges significantly from SPARF in both approach and outcomes, delivering superior performance on 3DGS. The depth consistency loss (DCons) employed in SPARF is fundamentally limited by its reliance on single-view projections, where depth inaccuracies lead to flawed projections. The DCons in SPARF's original paper improves PSNR by 0.20 (from 20.81 to 21.01) on NeRF—a result that aligns with our own reproduction on 3DGS. This proves that the single-view projection on SPARF is insufficient to address the overfitting issue on both NeRF and 3DGS. In contrast, our depth supervision for novel views in Table 4iii improves the PNSR by 3.13.
> > >
> > > ***Q2: Analysis for semantic loss***
> > >
> > > ***A2:*** As explained in Lines 271-277 and Lines 441-443, DINOv2 and CLIP down-sample features early in the encoder with a large stride, e.g. 16. It is widely recognized that features extracted with a large stride often result in a loss of detail, which is the reason why the semantic loss with DINOv2 and CLIP obtains poorer performance. Additionally, Figure 3 further illustrates that using DINOv2 and CLIP as feature extractors generates worse results in some boundary and detailed regions. In contrast, our semantic loss using VGG can reduce artifacts and maintain details.
> > >
> > > ***Q3: Details of SFM initialization***
> > >
> > > ***A3:*** We have already explained in our previous response that we initialize the 3D Gaussians with dense reconstruction in COLMAP. This process involves two parts, SfM followed by MVS, and does estimate camera extrinsics during SfM, therefore, the statement that "this process does not estimate any camera poses" is simply incorrect. Training and test views are both served as input for SfM to obtain their poses, then only the training views are used during MVS to get the fused point clouds. The technically sound initialization method name for the shorthand "SfM initialization" would be "Initialization based on the poses estimated by SfM for all views, and the fused point clouds from MVS on training views only", which will need to be shortened when referring to. Thus the initialization method we used is actually SfM+MVS, rather than just MVS as the reviewer suggested.
> > >
> > > The integrity of our initialization is by no means affected by the name, we ensure the dense point clouds are only from the training views, making it fair and comparable with the baselines.
> > >
> > > ***Q4: Comparison of training time and results***
> > >
> > > ***A4:*** Our method, which incorporates RoMa and VGG, achieves significantly reduced training time while delivering superior performance in sparse view synthesis compared to networks relying on pre-trained depth/diffusion models, such as FSGS and SparseGS. As noted in our previous responses, only the VGG model introduces additional training time. However, as detailed in Table S7 of the uploaded file, our method requires just 5.82 minutes of training—substantially less than the 51.78 minutes needed by SparseGS (which uses a pre-trained diffusion model) and the 10.90 minutes required by FSGS  (which relies on a pre-trained depth network).
> > >
> > > Furthermore, our approach not only enhances efficiency but also surpasses SparseGS and FSGS in performance, as demonstrated in Table 1. While incorporating VGG does involve some additional training time compared to raw 3DGS, the efficiency and speed trade-off is outweighed by the significant improvements in performance. Our method offers considerable gains over both raw 3DGS and methods reliant on pre-trained depth or diffusion models. For example, it improves PSNR by 4.09 dB on DTU and 3.32 dB on Blender compared to 3DGS.
> > >
> > > In summary, the integration of RoMa and VGG enhances the overall speed and effectiveness of our method, making it a notable advancement over previous approaches.

---

> > > > ### Comment · Reviewer_oLUf · 2024-08-14
> > > >
> > > > The authors alway passby the most important problems but only quibble by mentioning some irrelevant and trivial things, attempting to divert the attention. They can't face the exposed problems but only deny them. Such a response is meaningless. As above, this paper has many serious problems. The quality can not match NeurIPS. I'll lower my rating.
> > > >
> > > > BTW, for the  initialization part, the author  said that "We follow the SfM process provided in FSGS ('tools/colmap_llff.py' in FSGS code) to generate camera poses and 3D points". Thus, do the authors still claim that "This process involves two parts, SfM followed by MVS, and does estimate camera extrinsics during SfM, therefore, the statement that "this process does not estimate any camera poses" is simply incorrect"? See what 'tools/colmap_llff.py' exactly includes. Where is SfM in this file?

---

> > > > > ### Author Response · Authors · 2024-08-14
> > > > > **Reply to Reviewer oLUf**
> > > > >
> > > > > We sincerely appreciate your time and efforts in reviewing our responses. We hope that our responses below adequately address your concerns.
> > > > >
> > > > > We believe that the focus on specific terminology, rather than the actual process and results, is not conducive to the evaluation of the work's quality. We have been transparent in our methodology and provided a thorough explanation in our previous responses (we initialize the 3D Gaussians with dense reconstruction in COLMAP.). This step is performed within the COLMAP framework, specifically using the convert.py script in our provided FSGS code to extract the camera poses.
> > > > >
> > > > > Our use of SfM followed by MVS is technically accurate and appropriately implemented in the context of our research.

---

### Official Review · Reviewer_ZPEG · 2024-07-08

**Soundness:** 3
**Presentation:** 4
**Contribution:** 3
**Rating:** 6
**Confidence:** 4

**Summary:**

This paper introduces a novel few-shot Gaussian Splatting method for synthesizing novel views. Unlike conventional approaches that rely on pre-trained monocular depth estimation or diffusion methods, the proposed method leverages the matches of available training views to generate novel sample views between the training frames. It employs color, depth, and image feature losses. Additionally, a novel regularization loss is introduced to preserve the local structure of the object. Experimental results demonstrate that the proposed method achieves significant performance improvements in both real and synthetic datasets.

**Strengths:**

The strengths of the paper are as follows:

1. The proposed few-shot Gaussian splatting method for novel view synthesis, which does not rely on pre-trained depth estimation or diffusion models while achieving state-of-the-art performance. It is important to note that pre-trained depth estimation and diffusion models often require large parameters, which can lead to longer training times.
2. A multi-stage training scheme consisting of pre-training, intermediate, and tuning stages. This scheme optimizes the scene representation gradually. The pre-training process aims to obtain a basic representation of the scene using known views. The intermediate process transfers knowledge from known views to novel views while preserving consistency in overlapping regions between known training views. Finally, the tuning process removes artifacts that occur in few-shot scenarios.
3. The consistency loss function, which maintains the similarity of color, depth, and image features between pixels in novel views projected from known training views. This ensures that novel views have similar semantic, color, and depth information. It is worth noting that the loss functions are adaptively weighted to minimize the impact of errors in texture-rich regions.
4. The locality loss function, which maintains color similarity between the Gaussian and its neighboring regions. Accurate rendering results are more likely to occur when color values are smooth between neighborhoods.
5. The paper provides an exhaustive ablation study for each network design decision, leading to a convincing algorithm design.

**Weaknesses:**

The weaknesses of the paper are as follows:

1. The authors claim that relying on depth estimation or diffusion priors requires longer training times, but there is no comparison or ablation study provided to justify this claim. It would be beneficial to perform a comparison between state-of-the-art methods and the proposed method in terms of training and inference time.
2. The paper does not provide a justification for why the proposed method, which relies on matches between training views, performs better than methods that rely on pre-trained depth estimation or diffusion methods. A more in-depth analysis is needed to demonstrate how the proposed method outperforms these approaches.

**Questions:**

Based on the weaknesses, the following questions arise:

1. How do the training and inference times of the proposed method compare to those of state-of-the-art methods?
2. How well does the performance of the proposed method generalize to unseen data? Given that state-of-the-art methods typically employ large pre-trained models for data reconstruction, the proposed method, which relies solely on matches between known views, may encounter difficulties when dealing with unseen regions.

**Limitations:**

The proposed method may face challenges when dealing with texture-rich regions that are not visible from the input views.

---

> ### Author Rebuttal · Authors · 2024-08-07
>
> We thank the reviewer for the insightful and thorough review. We will integrate the additional results and analysis in the next version. In the following, we address the main concerns raised in this review.
>
> ***Q: How do the training and inference times of the proposed method compare to those of state-of-the-art methods?***
>
> ***A:*** Thank you for your valuable feedback on the efficiency evaluation. Table S7 of the uploaded file compares the training time of 3DGS-based methods. Our method demonstrates comparable training time and is faster than SparseGS [41] (which uses a pre-trained diffusion model) and FSGS [49] (which relies on a pre-trained depth network). For inference time, since all methods utilize the same rendering process as 3DGS, their inference time is similar across these methods, at around 300 FPS.
>
> ***Q: The paper does not provide a justification for why the proposed method, which relies on matches between training views, performs better than methods that rely on pre-trained depth estimation or diffusion methods. A more in-depth analysis is needed to demonstrate how the proposed method outperforms these approaches.***
>
> ***A:*** We appreciate the reviewer for the valuable suggestions and will incorporate the following analysis in the updated version. Our method offers several advantages over existing approaches, contributing to its improved performance: (1) **Address overfitting**: Methods that incorporate pre-trained depth estimation networks, such as DNGaussian [16] and DRGS [5], add depth supervision only on known view, which does not solve the issue of overfitting to sparse training view. In contrast, our method designs extra appearance/geometry/semantics constraints on novel views, which can address overfitting and enhance performance. (2) **Provide accurate supervision**: The methods, such as FSGS [49] and SparseGS [41], rely on a pre-trained depth network or diffusion model to generate pseudo labels for supervising the novel view. This may generate inaccurate pseudo labels, causing noises and encountering scale-ambiguity issues with the generated depth. Besides, the diffusion model used in SparseGS increases the training time significantly (see Table S7 of the uploaded file). Contrastively, with feature matching, our method projects the pixels from two training views to the novel view to provide supervision. This can leverage the ground truth of color from the training view, which is more accurate than pseudo labels generated by other networks, and ensure consistency across multiple views, thereby enhancing overall performance.
>
> ***Q: How well does the performance of the proposed method generalize to unseen data?***
>
> ***A:*** In the orange boxes of Figure S1 (see the uploaded file), the visualization of unseen regions demonstrates that our method yields fewer artifacts and superior performance than FSGS [49] (which uses a pre-trained depth network). This can be attributed to that our multi-view consistency constraints can provide accurate supervision for the seen regions, and the proposed locality preserving regularization helps maintain local smoothness and reduce artifacts.

---

### Official Review · Reviewer_8CAh · 2024-07-13

**Soundness:** 3
**Presentation:** 3
**Contribution:** 2
**Rating:** 5
**Confidence:** 5

**Summary:**

This paper proposes a new method for sparse-view novel view synthesis. It proposes a multi-stage training scheme including pre-training, intermediate, and tuning stages. It introduces pre-trained dense matching models to find pixel correspondences between different-view images and encourage consistency. A Locality Preserving Regularization is proposed to encourage local smoothness.

**Strengths:**

1. The proposed method achieves SOTA performances on the 3-view LLFF dataset, 3-view DTU dataset and 8-view Blender dataset.
2. The proposed method suits both random points and mvs points as shown in Table 1.
3. The paper is well-organized.

**Weaknesses:**

1. The comparisons are not enough. It lacks comparisons on more input views, such as the 6-view and 9-view settings used in FreeNeRF, which is also important to evaluate the proposed method in sparse-view settings.
2. The ablation studies are not enough. There are lots of hyperparameters listed in the implementation details, however, it is unclear how these hyperparameters are selected and how they impact the performance, which is important to evaluate the robustness of the proposed method.
3. The proposed method introduces ore-trained dense matching networks, thus I think it is still similar to those introduced pre-trained depth estimation networks.

**Questions:**

1. I wonder is the proposed method still work with more input views such as 6 views and 9 views.

**Limitations:**

The authors have discussed the limitations.

---

> ### Author Rebuttal · Authors · 2024-08-07
>
> We thank the reviewer for the insightful and thorough review. We will integrate the additional results and analysis in the revised version. In the following, we address the main concerns raised in this review.
> ***Q: It lacks comparisons on more input views, such as the 6-view and 9-view settings used in FreeNeRF.***
>
> ***A:*** We thank the reviewer for the constructive suggestions on experiments. The results for 6-view and 9-view settings are given in Table S1 of the uploaded file. Our method shows improved performance over the baseline 3DGS with both SFM and random initialization. Notably, with SFM initialization, our method surpasses FreeNeRF on both datasets. We will include these results in the revised version.
>
> ***Q: It is unclear how these hyperparameters are selected and how they impact the performance.***
>
> ***A:*** We appreciate the reviewer's suggestions on the ablation study. We have given evaluations for hyperparameters $\alpha, \beta, \gamma, \eta$, and the multi-stage settings in Table S2 - Table S6 of the uploaded file on DTU with 3 training views. (1) **$\alpha$ in Eq 12**: we set it to a low weight due to the lack of ground truth for the depth. It can be seen that, in Table S3, larger $\alpha$ (e.g. 0.2) gets poorer results, primarily due to noise in predicted depth values. (2) **$\beta$ in Eq 12**: due to the importance of color supervision for the novel view synthesis, a large weight is utilized in the experiment. In Table S2, a value of 0.5 achieves the best performance. (3) **$\gamma$ in Eq 12**: The results in Table S5 demonstrate that our method is robust to different $\gamma$ values, with $\gamma=0.001$ being optimal. (4) **$\eta$ in Eq 15**: The pre-training loss in the intermediate stage is employed to ensure sufficient supervision for novel views. A lower $\eta$ (0.01) does not solve this very well, while higher $\eta$ (0.1 and 0.5) fails to prevent overfitting to the training views. (5) **Multi-stage settings**: we evaluate the influence of iterations for each stage in our 3-stage training in Table S6. The last two rows show that a fewer/more iteration in the first stage leads to worse results, due to underfitting/overfitting to training views. The 1st-4th rows demonstrate that the fine-tuning stage enhances performance and achieves optimal results with 500 iterations. We will integrate these results and analysis to our revised version.
>
> ***Q: The proposed method introduces pre-trained dense matching networks, thus I think it is still similar to those introduced pre-trained depth estimation networks.***
>
> ***A:*** Our method offers several advantages over existing approaches with a pre-trained depth estimation network, contributing to its improved performance: (1) **Address overfitting**: Methods that incorporate pre-trained depth estimation networks, such as DNGaussian [16] and DRGS [5], include depth supervision only on known view, which does not solve the issue of overfitting to sparse training view. In contrast, our method designs extra appearance/geometry/semantics constraints on novel views, which can avoid overfitting and enhance performance. (2) **Provide accurate supervision**: The methods, such as FSGS [49], rely on a pre-trained depth network to generate pseudo labels for supervising the novel view. This may generate inaccurate pseudo labels, encountering scale-ambiguity issues with the generated depth. Contrastively, with feature matching, our method projects the pixels from two training views to the novel view to provide supervision.  This can leverage the ground truth of color from the training view, which is more accurate than pseudo labels generated by other networks, and ensure consistency across multiple views, thereby enhancing overall performance.

---

### Official Review · Reviewer_Eqzk · 2024-07-13

**Soundness:** 3
**Presentation:** 3
**Contribution:** 3
**Rating:** 6
**Confidence:** 4

**Summary:**

This paper tackles the problem of few view (or sparse view) 3DGS multistage training with correspondence-driven losses that enforce projected colors, depths, and semantic features (extracted by a pre-trained VGG) are consistent. Contributions are straightforward and geometrically inspired.

In addition, the authors propose a locality preservation loss, which enforces color smoothness in neighbouring gaussians.

**Strengths:**

1. Simple yet effective geometrically inspired regularization for few view 3DGS.
2. Strong qualitative and quantitative results.
3. Extensive ablation studies.

**Weaknesses:**

1. Intuitions behind a 3-stage training strategy are not well established. If intermediate training already includes L_pre_training, why is further fine-tuning with L_pre_training only needed?

**Questions:**

See weaknesses

**Limitations:**

Ok

---

> ### Author Rebuttal · Authors · 2024-08-07
>
> We appreciate the reviewer’s insightful and thorough review. We will further integrate the explanation in the revised version. Below, we address the main concerns raised in this review.
>
> ***Q: Intuitions behind a 3-stage training strategy are not well established. If intermediate training already includes L_pre_training, why is further fine-tuning with L_pre_training only needed?***
>
> ***A:*** The design of 3 stages aims to address overfitting to sparse training views and improve supervision reliability in novel views. Specifically, (1) the pre-training stage aims to obtain a basic representation of the 3D scene. (2) The intermediate stage mainly focuses on integrating additional view information into the network. We found that the sparse supervision and noises of pseudo labels in the novel views cause model collapse in some scenes. Thus, $L_{pre-training}$ is applied in this stage to solve this issue, employing a lower weight to prevent overfitting to the training view. (3) The fine-tuning stage is designed to refine the network with ground truth, add supervision for unmatched pixels during the intermediate stage, and minimize the impact of noise in the novel view. The results in the 1st-4th row of Table S6 (see the uploaded file) also indicate that incorporating this extra fine-tuning stage improves performance. We will include these explanations to our revised version.

---

> > ### Comment · Reviewer_Eqzk · 2024-08-09
> > **Thanks for your reply**
> >
> > Thanks for supplementing your results. It is now clear the 3rd stage provides additional performance improvements. Is is possible to show those "unmatched pixels during  the intermediate stage" to support your claims?
> >
> > I am also wondering why you did not include an experiment with 1st: 2000 2nd: 7500 Third: 0, for a more direct comparison.

---

> ### Author Response · Authors · 2024-08-10
> **Reply to Reviewer Eqzk**
>
> We sincerely appreciate your precious time and efforts in reviewing our paper and responses. We hope that the response provided below adequately addresses your concerns.
>
> ***Q1: Is it possible to show those "unmatched pixels during the intermediate stage" to support your claims?***
>
> ***A1:*** We appreciate the review's suggestion on visualizations to better support our claim. We provided visualizations for the matched results in Figure 7. The white regions in Figure 7c and the areas without red lines in Figure 7d represent unmatched pixels, which lack effective supervision during the intermediate stage. We will include detailed descriptions in our revised version.
>
> ***Q2: I am also wondering why you did not include an experiment with 1st: 2000 2nd: 7500 Third: 0, for a more direct comparison.***
>
> ***A2:*** We thank the reviewer for the suggestion on the experiment. The setting with '1st: 2000 2nd: 7500 Third: 0' achieves a performance of PSNR 18.37, SSIM 0.782, and LPIPS 0.198. We will integrate these results into our revised version.

---

### Author Rebuttal · Authors · 2024-08-07

We appreciate the constructive feedback and positive comments from the reviewers. We are encouraged that the reviewers found that:
- Our paper is well-organized (Reviewer 8CAh).
- Our method is effective and convincing (Reviewer Eqzk, Reviewer ZPEG).
- Our method achieves SOTA performance, with strong qualitative and quantitative results, as well as extensive ablation studies (Reviewer Eqzk, Reviewer 8CAh, Reviewer ZPEG, Reviewer oLUf).
***
We have uploaded a PDF file that includes two figures and several tables to address the reviewers' concerns:
- Figure S1: Visualizations of unseen regions generated by our method and FSGS.
- Figure S2: Qualitative depth comparison between the baseline and our method.
- Table S1: Results for 6-view and 9-view settings.
- Table S2 - Table S6: Evaluation of the influence of hyperparameters.
- Table S7: Comparison of the training time for 3DGS-based methods in few-shot novel view synthesis.
***
We address the reviewers' questions below, and will incorporate the suggested results and description into our revised version. We will actively participate in the Author-Reviewer discussion session. Please feel free to tell us if anything remains unclear.
***
We sincerely appreciate PCs, ACs, and reviewers for their time and effort in evaluating our submission.

---

### Decision · Program_Chairs · 2024-09-25

**Decision:**

Accept (poster)

**Comment:**

This paper proposes a method for 3D Gaussian reconstruction from sparse view images. The framework includes three stages, where the first stage initializes 3D Gaussians using sparse views, the second stage synthesizes novel views with multi-view consistency constraint, the last stage further refines the scene for texture improvement.

During rebuttal, reviewers raised concerns about the motivation of 3-stage design (Eqzk), performance on 6/9-views (8CAh), hyper-parameters (8CAh, oLUf), novelty (8CAh, ZPEG, oLUf) and initialization. The authors have provided additional experiments for 6/9-view setting, ablation studies on hyper-parameters, as well as differentiating with existing methods that utilize depth prior. For the discussion on initialization, the method follows standard dataset pre-processing (i.e., predicting camera poses using all views, and point cloud from training views only). This initialization facilitates a fair comparison with existing baseline methods.

Based on these consideration, this paper provides useful insights as well as effective designs for sparse-view reconstruction and is recommended for acceptance. The authors are encouraged to include additional experiments and ablation studies in the revision.